# FT-IR Spectral Signature of Sensitive and Multidrug-Resistant Osteosarcoma Cell-Derived Extracellular Nanovesicles

**DOI:** 10.3390/cells11050778

**Published:** 2022-02-23

**Authors:** Francesca Perut, Gabriela Graziani, Laura Roncuzzi, Nicoletta Zini, Sofia Avnet, Nicola Baldini

**Affiliations:** 1Biomedical Science and Technologies Lab, IRCCS Istituto Ortopedico Rizzoli, Via di Barbiano 1/10, 40136 Bologna, Italy; laura.roncuzzi@ior.it (L.R.); nicola.baldini@ior.it (N.B.); 2Laboratory of Nanobiotechnology, IRCCS Istituto Ortopedico Rizzoli, Via di Barbiano 1/10, 40136 Bologna, Italy; gabriela.graziani@ior.it; 3CNR Institute of Molecular Genetics “Luigi Luca Cavalli-Sforza”, Unit of Bologna, 40100 Bologna, Italy; nicolettazini@gmail.com; 4IRCCS Istituto Ortopedico Rizzoli, 40136 Bologna, Italy; 5Department of Biomedical and Neuromotor Sciences, University of Bologna, Via di Barbiano 1/10, 40136 Bologna, Italy; sofia.avnet@ior.it

**Keywords:** extracellular nanovesicles, osteosarcoma, multidrug resistance, FT-IR/ATR spectroscopy, non-invasive diagnostics

## Abstract

Osteosarcoma (OS) is the most common primary bone cancer in children and adolescents. Despite aggressive treatment regimens, the outcome is unsatisfactory, and multidrug resistance (MDR) is a pivotal process in OS treatment failure. OS-derived extracellular vesicles (EVs) promote drug resistance to chemotherapy and target therapy through different mechanisms. The aim of this study was to identify subpopulations of osteosarcoma-EVs by Fourier transform infrared spectroscopy (FT-IR) to define a specific spectral signature for sensitive and multidrug-resistant OS-derived EVs. EVs were isolated from sensitive and MDR OS cells as well as from mesenchymal stem cells by differential centrifugation and ultracentrifugation. EVs size, morphology and protein expression were characterized. FT-IR/ATR of EVs spectra were acquired in the region of 400–4000 cm^−1^ (resolution 4 cm^−1^, 128 scans). The FT-IR spectra obtained were consistently different in the EVs compared to cells from which they originate. A specific spectral signature, characterized by a shift and a new band (1601 cm^−1^), permitted to clearly distinguish EVs isolated by sensitive and multidrug-resistant OS cells. Our data suggest that FT-IR spectroscopy allows to characterize and define a specific spectral signature for sensitive and MDR OS-derived EVs.

## 1. Introduction

Osteosarcoma (OS) is the most common primary bone tumor, mainly occurring in children, teens, and young adults [1]. The 5-year survival rate for localized osteosarcoma is about 70%, however, despite aggressive treatments, it is dramatically lower (20%) in patients with metastases at diagnosis or recurrent disease [2]. Chemoresistance is still a major cause of relapse in OS and has been associated with impaired drug transport or increased drug efflux due to the overexpression of efflux transporters such as the P-glycoprotein, encoded by the multidrug resistance protein 1 gene (MDR-1) [3,4]. Additional factors also play a role in the natural history and the responsiveness to chemotherapy of OS. Among these, the tumor microenvironment and cell-to-cell interactions are well-known regulators of the tumorigenic process and contribute to cancer progression. The multidirectional and dynamic interactions among cells are mediated by soluble factors and extracellular vesicles (EVs). Indeed, EVs and their cargo have been associated with OS growth, metastasis, and chemoresistance [5,6,7,8]. In fact, we have previously demonstrated that multidrug-resistant OS cells are able to spread their ability to resist the effects of doxorubicin to sensitive cells by transferring EVs carrying P-glycoprotein and MDR-1 mRNA [6]. Moreover, EVs can contribute to exacerbating drug resistance by additional mechanisms. EVs can directly interact and sequest drugs or transfer drug resistant phenotype to sensitive cells by changing their transcriptome [9,10]. The possibility to distinguish EVs derived from sensitive and multidrug-resistant OS cells may help to follow OS progression, and potentially improve the effectiveness of conventional therapies through personalized medicine approaches.

Fourier-transform infrared spectroscopy (FT-IR) has been fruitfully applied to describe structural characterization of proteins and protein–membrane interactions in biological matrices [11]. Malignant cell characteristics and cell differentiation show specific vibrational molecular signatures, thus indicating a modified chemical composition associated with biomolecules such as proteins, nucleic acids, lipids, and carbohydrates [12,13,14]. Vibrational spectroscopy is simple, label-free, sensitive, and requires minimal sample preparation. Therefore, FT-IR spectroscopy could also be applied to characterize EVs and the cells from which they derive. Indeed, Mihály et al. have been able to distinguish exosomes, microvesicles, and apoptotic bodies based on specific FT-IR profiles derived from amide and CH stretching vibrations [15]. Specific fingerprints of EVs subpopulations (i.e., large (~600 nm), medium (~200 nm), and small (~60 nm EVs)) from prostate cancer and melanoma cell lines have been identified by FT-IR analyses [16].

In this study, we challenged FT-IR spectroscopy to fingerprint OS-derived EVs to define a specific spectral signature for sensitive and multidrug-resistant OS-derived EVs. For this purpose, we purified and characterized EVs from doxorubicin-sensitive and resistant OS cells and mesenchymal stromal cells (MSC). We identified different spectrum profiles for EVs and for the cells from which they derive. Moreover, we identified a specific pattern, characterized by a shift and a new band, which permitted to clearly distinguish EVs isolated from sensitive and doxorubicin-resistant OS cells. These results demonstrated that FT-IR spectroscopy is a non-invasive method that should be further investigated for EV characterization and can be utilized in EV-based diagnostic and prognostic approaches.

## 2. Materials and Methods

### 2.1. Cell Culture

The human OS cell lines MG-63 and 143B were purchased from the American Type Culture Collection (ATCC, Manassas, VA, USA). The MDR cell line MG-63DXR30 was established from the parental MG-63 [17]. MG-63 and 143B cells were maintained in Iscove’s Modified Dulbecco’s Medium (IMDM, Invitrogen, Carlsbad, CA, USA), supplemented with 10% heat-inactivated fetal bovine serum (FBS), penicillin (100 U/mL), and streptomycin (100 µg/mL) (Sigma-Aldrich, Milan, Italy). Drug-resistant variant MG-63DXR30 was continuously cultured in the presence of the selective drug concentration (30 ng/mL doxorubicin), except when the supernatant was collected to isolate EVs. Adipose-derived mesenchymal stem cells (ADMSCs) were purchased from the ATCC, and they were grown in α-minimum essential medium (α-MEM) supplemented with 10% FBS, 100 U/mL penicillin, and 100 µg/mL streptomycin). Passage 4–5 ADMSCs were used in all the experiments. Cells were maintained at 37 °C in a humidified 5% CO_2_ atmosphere, and periodically tested for mycoplasma contamination.

### 2.2. Extracellular Nanovesicles (EVs) Isolation and Purification

Cells were cultured until 70–80% confluence, then washed with phosphate-buffered saline (PBS) and incubated for two consecutive periods (18 h and additional 18 h) with IMDM supplemented with 10% FBS depleted of extracellular nanovesicles obtained via ultracentrifugation [18]. Cell density and viability were assessed by the erythrosine B (Sigma-Aldrich) dye exclusion method [19]. EVs pellet was obtained from the supernatant collected from MG-63, MG-63DXR30, 143B, and ADMSC cells grown on 15 Petri dishes (diameter 150 mm, 18 mL/Petri). The EVs were concentrated by a series of differential centrifugations: 500× *g* for 10 min (twice), 2000× *g* for 15 min (twice), and 10,000× *g* for 30 min (twice) at 4 °C to remove floating cells and cellular debris. The supernatant was then ultracentrifuged at 110,000× *g* for 1 h at 4 °C (Beckman Coulter, Milan, Italy). The EVs pellet was washed (110,000× *g* for 1 h at 4 °C), resuspended in PBS, and stored at −80 °C until use. EVs quantity was determined by the Bradford method (Bio-Rad, Milan, Italy). The EVs derived from the supernatant of MG-63 and from the medium of MG-63DXR30 were named EVs/s and EVs/DXR, respectively.

### 2.3. Transmission Electron Microscopy

EVs were resuspended in 2% paraformaldehyde (PFA) and loaded onto formvar-carbon coated grids (Electron Microscopy Sciences, Hatfield, PA, USA). After fixation in 1% glutaraldehyde, EVs were washed, counterstained with a solution of uranyl oxalate (pH 7.0), and embedded in a mixture of 4% uranyl acetate and 2% methylcellulose. EVs were observed with a Zeiss-EM 109 transmission electron microscope (Zeiss, Oberkochen, Germany). Images were captured by using the NIKON digital camera Dmx 1200F, and ACT-1 software (NIKON Corporation, Tokyo, Japan). The EVs diameter was measured (n > 200), and the percentage of size distribution was calculated.

### 2.4. Western Blot Analysis

EVs and cell pellets were lysed with RIPA buffer (25 mM Tris-HCl pH 7.6, 150 mM NaCl, 1% NP-40, 1% Na-deoxycholate, 0.1% SDS) added with protease inhibitor cocktail (Roche, Milan, Italy) (30 min at 4 °C). Nuclei and cell debris were removed by centrifugation. The protein concentration was determined by using the Bradford assay. EVs and total cellular proteins were resolved by 10% SDS-polyacrylamide gel and transferred to a nitrocellulose membrane (Thermo Fisher Scientific, Waltham, MA, USA). After blocking with 5% dry milk (Thermo Fisher Scientific) in T-TBS (0.1 M Tris-HCl pH 8.0, 1.5 M NaCl, and 1% Tween-20) for 1 h at room temperature, the membranes were incubated with rabbit polyclonal CD9, CD63, CD81, and hsp70 (1:1000) (System Biosciences, Palo Alto, CA, USA) antibodies or mouse polyclonal anti-calnexin (1:500) (sc-23954, Santa Cruz) antibody overnight at 4 °C. After washing in 0.05% Tween-20 in PBS, the membranes were incubated with goat anti-rabbit antibody (EXOAB kit) (1:20,000) or goat anti-mouse antibody (nif 825, Amersham, Ge Healthcare, Freiburg, Germany) (1:1000) conjugated to horseradish peroxidase for 1 h at room temperature. Immunocomplexes were detected with the ECL western blot analysis system (Euroclone, Milan, Italy). Reversible Ponceau S (Sigma-Aldrich) staining was used to assess equal gel loading.

### 2.5. Fourier Transform Infrared Analysis

Cell pellets and EVs were resuspended in PBS to avoid possible inference of the bands of crystallized medium in the 400–1200 cm^−1^ area. Bands of PBS were acquired at each time point, following the same conditions used for EVs samples. The acquisition was carried out on EVs (5 µL containing 9 ± 2 µg total protein) or cell pellet (5 µL containing 0.5 × 10^6^ cells)-containing droplets, left to dry on the ATR crystal. For each acquisition, 5 µL EVs or cell pellet in PBS were used and deposited on the crystal. The acquisition was performed by a Perkin Elmer Spectrum 2 instrument, equipped with a diamond crystal ATR, with the following acquisition parameters: acquisition range 4000–400 cm^−1^, resolution 4 cm^−1^, 128 scans, step size 0.5 cm^−1^. For EV data processing, all bands were scaled with respect to the external band of PBS at 1084 cm^−1^, and then the PBS curve was subtracted from each curve. These curves gave a first indication of the total amount of proteins and lipids contained in the samples. To compare the protein: lipid ratio, all curves were scaled with respect to the Amide I band at 1650 cm^−1^. These data permitted to evaluate the variability among different cells types as well as the internal variability within each group.

### 2.6. Statistical Analysis

Statistical analysis was performed using the Graph Pad Prism 7.04 software for Windows (Graph Pad Software, La Jolla, CA, USA). Results were reported as the mean ± standard deviation and differences between groups were analyzed by using the non-parametric Mann–Whitney test. Only *p* < 0.05 were considered significant.

## 3. Results

### 3.1. Extracellular Nanovesicles Isolation and Characterization

Extracellular nanovesicles isolated from sensitive and doxorubicin-resistant OS cells were morphologically homogeneous with a typical round or cup-shaped appearance, as shown by transmission electron microscopy analysis (Figure 1a). The size distribution of OS-derived EVs is described in Figure 1b. Extracellular nanovesicle purity was verified by Western blotting analysis for transmembrane and cytosolic proteins typically recovered in EVs [20]. As shown in Figure 1c, CD9 and CD63 were strongly enriched in OS-derived EVs preparations compared to cell lysates and the heat-shock protein 70 (hsp70) was expressed in all samples. EVs preparations were negative for the endoplasmic reticulum protein calnexin. The amount of EVs released by doxorubicin-resistant cells was significantly higher than the amount of EVs derived from OS-sensitive cells (*p* = 0.004) (Figure 1d).

### 3.2. Analysis of Fourier-Transform Infrared (FT-IR) Spectra of EVs and Cells of Origin

FT-IR spectroscopy simultaneously provides information on different biomolecules contained in a biological system, such as lipids, proteins, nucleic acids, and carbohydrates. The spectroscopy region most used for biological applications is the mid-infrared in the 2.5–25 μm (4000–400 cm^−1^) [21].

As both EVs and cells were resuspended in PBS before acquisition, the IR spectrum of PBS, as control, is reported in the Appendix A, showing different IR features in comparison with those ascribed to EVs and cells. The PBS spectrum showed the characteristic sharp bands at 1076, 976, 857 cm^−1^ (ν P-O) and broad bands at 1660 and ~3000 cm^−1^ [22]. Proper subtraction of the band at 1660 cm^−1^ was essential, as it overlapped with the Amide I band. No differences were registered in the PBS spectra acquired at different time points, which, instead, perfectly overlap.

FT-IR spectra of EVs derived from OS sensitive and doxorubicin-resistant cells are shown in Figure 2. The analysis of FT-IR spectra of EVs revealed specific bands. In particular, all EVs curves showed the amide I absorption band located at ≈1649 cm^−1^, associated with the C=O stretching mode of the peptide bond, and the amide II absorption located at ≈1455 cm^−1^, which was primarily ascribable to N-H vibrations of peptide groups. A band is also assessed at 1407 cm^−1^, which can be attributed to symmetric and asymmetric vibration of COO− [23].

However, bending in the 1400 cm^−1^ area is related both to lipids and to proteins, so different contributions might impact the band at 1407 [16]. Additionally, the absorption of the ester groups of phospholipids, triglycerides, and cholesterol esters was revealed at 1738 cm^−1^, and acyl group vibrations appeared at 2930 and 2852 cm^−1^.

A band on the interval at 986–992 cm^−1^ (986 cm^−1^) was associated with the ribose phosphate main chain, whereas the band at 966 cm^−1^ (absent here) arose from the stretching vibration of the DNA backbone [13]. The spectra of nucleic acids are characterized in four spectral regions: the region of 1780–1550 cm^−1^ for in-plane vibrations of double bonds of the bases, the region of 1550–1270 cm^−1^ for the deformation vibrations of the bases coupled with the sugar vibrations, the region of 1270–1000 cm^−1^ for vibrations of –PO2−, and the region of 1000–780 cm^−1^ for the vibrations of the sugar-phosphate backbone. Here, because of the overlapping of several bands, only the band at 1318 was clearly visible. The intra-sample repeatability of FT-IR spectra was high, as shown in the Appendix A for MG-63-derived EVs replicate acquisitions.

Comparing FT-IR profiles of EVs and the OS cells from which they were isolated, we could observe the same profile in the part of the spectra above 1300 cm^−1^, with the same bands revealed for cells and EVs, although with different intensity and variability (Figure 3a). However, EVs and cells showed a completely distinct profile in the 900–1300 cm^−1^ area, which made each EV clearly distinguishable from the relevant cells. This region has been linked to the stretching vibration of the nucleic acid backbone [13]. Indeed, cells showed bands at 1240 and 1040 cm^−1^, which were not detected in the EVs, and a relevant increase in the intensity of the band at 1080 cm^−1^. Bands at 1240 and 1080 cm^−1^ were due to the symmetric and asymmetric stretching modes of the phosphodiester groups, so they depended on the nucleic acid content of the cell. The band at 1040, instead, was associated with polysaccharides [24,25,26]. In addition, while OS cells showed a very low variability among replicates and sensitive/non-sensitive cell types, (all curves essentially overlapped in the protein and lipid zones), higher intra-specimen variability was assessed for the EVs, together with significant differences among the different EV groups (Figure 3b–d).

After normalization to equal intensities with respect to the Amide I band at 1649 cm^−1^, we compared the FT-IR spectra of different EV samples. FT-IR spectral differences between MG-63-derived EVs and MG-63DXR30-derived EVs were observed in the band regions of 1650–1545 cm^−1^. Indeed, the band at 1601 cm^−1^ appeared only in MG-63-derived EVs and a shift in the bands at 1649 cm^−1^ was found (Figure 4a–c). The presence of the peculiar band at 1601 cm^−1^ was also confirmed in EVs derived from the chemosensitive OS cell line 143B, which we have previously characterized [27] (Appendix A). In addition, a much higher variability was observed in the quantity of lipids compared to proteins for doxorubicin-resistant samples. Protein organization was similar in EVs derived from sensitive and doxorubicin-resistant OS cells. Spectroscopic protein-to-lipid ratio verified by comparing the intensity of the relevant bands was variable in MG63DXR30-derived EVs while remaining almost constant into MG-63-derived EVs.

Additionally, we analyzed spectra profiles of mesenchymal stromal cells and MSC-derived EVs. Transmission electron microscopy analysis showed that the extracellular nanovesicles isolated from MSC cells were morphologically homogeneous with a typical round- or cup-shaped appearance and were characterized for the tetraspanins CD63, CD9, CD81, and cytoplasmic hsp70 expression (Appendix A).

MSC and OS cell spectra profiles were similar, while OS and MSC-derived EV profiles and spectroscopic protein-to-lipid ratio were completely different (Figure 5). Indeed, although the position of the bands is the same, the intensity of those in the lipids zone is relevantly higher, indicating an important shift in the protein:lipid ratio.

## 4. Discussion

Fourier-transform infrared spectroscopy has recently been proposed to define healthy vs. disease profiles of cells and cell-derived vesicles. In fact, this technique is able to detect subtle changes in the molecular structure of nucleic acids, lipids, proteins, and carbohydrates in biological samples, thus defining a specific biomolecule fingerprinting. FT-IR spectroscopy has been utilized to identify and distinguish subpopulations of EVs derived from melanoma cells with different malignant grades [28], and it has been fruitfully applied to reveal differences at the single vesicle level between EVs derived from colon normal epithelial cells and colon cancer cells [23]. Moreover, FT-IR spectra were used to characterize EVs isolated from biological fluids and consistently different spectral signatures were identified for salivary cancer and healthy individual derived-EVs [29], for blood from prostate cancer and healthy patient derived-EVs [30], and for blood derived-EVs isolated from Alzheimer’s disease affected and control subjects [31].

In this study, we focused on the possibility to find a “biomolecular fingerprinting” based on FT-IR spectroscopy with the aim to distinguish EVs isolated from sensitive and drug-resistant osteosarcoma cells.

Extracellular vesicles secreted by drug-resistant cells are considered players of drug resistance maintenance and transfer in different types of tumors, including breast, prostate, colon, lung, gastric cancer as well as osteosarcoma [5,32]. This activity has been associated with specific EV cargo or EV properties. Indeed, breast cancer-derived EVs favor drug resistance by transferring pro-survival signals, reducing the intracellular accumulation of drugs, upregulating P-glycoprotein expression in sensitive cancer cells, and altering the epithelial–mesenchymal transition [33]. Similar mechanisms were also found in prostate cancer, where EVs contributed to docetaxel resistance [34], and ovarian cancer where EVs can deliver vascular-endothelial growth factor (VEGF) into endothelial cells causing resistance to anti-VEGF therapies [35]. EVs derived from the plasma of patients with acute myeloid leukemia (AML) confer idarubicin resistance in sensitive AML by inducing the expression of the drug efflux pumps MRD1 and MRP1 [36]. In stress conditions, nasopharyngeal carcinoma cells produced EVs containing the endoplasmic reticulum resident protein 44, which could be transferred to adjacent cells strengthening cisplatin resistance [37]. Additionally, a differential expression of exosomal circRNAs has been found in drug-resistant colon cancer cells compared to sensitive ones [38]. Furthermore, cells of the tumor microenvironment can be involved in chemoresistance by means of EVs cargo. Indeed, cancer-associated fibroblast-derived EVs are able to confer cisplatin resistance to non-small cell lung cancer [39] and macrophage-derived EVs mediated doxorubicin resistance in gastric cancer [40].

Unfavorable prognosis in OS is often associated with inherent or acquired drug resistance [41], a detrimental process in which EVs have shown a specific involvement. Indeed, it has been demonstrated that doxorubicin and cisplatin resistance are transferred from OS resistant to sensitive cells by means of EVs carrying P-glycoprotein, MDR-1 mRNA, or the circular RNA hsa_circ_103801 [6,42]. The study of the alterations in protein and RNA content of EVs secreted by drug-resistant cells and the isolation of circulating EVs may become a promising approach to discover and validate new biomarkers that could be used to improve disease monitoring.

The aim of the present study was to identify a specific FT-IR spectral signature for sensitive and drug-resistant OS-derived EVs. Thus, we purified EVs from doxorubicin-sensitive and resistant OS cells, and verified EV size and morphology by transmission electron microscopy. The strong enrichment in CD9 and CD63 proteins in EVs samples, compared to the cells from which they derive, and the absence of calnexin, an endoplasmic reticulum protein, confirmed the purity of extracellular vesicles samples [20]. The size distribution profile of isolated EVs was observed to be in the range of 50 to 200 nm, consistent with previously published reports [43,44]. In addition, according to results observed in ovarian and prostate cancer, we verified that drug-resistant OS cells release significantly higher amounts of EVs compared to the respective sensitive cells [45,46,47]. Increased EV shedding has also been reported when cells are subjected to hypoxia and acidic stress, conditions that characterize tumor mass persistence and aggressiveness [48,49].

FT-IR analysis was therefore applied to identify specific spectral fingerprints in EVs derived from OS-sensitive and drug-resistant cells. We analyzed spectral regions characteristic of the absorption bands of nucleic acids, lipids, proteins, and carbohydrates, as biological features associated with all cells and EVs, after appropriate subtraction of the PBS profile. Although the spectra were very complex and resulted from the overlapping absorption of multiple biomolecules, the absorption of the amide I, amide II, and ester groups of phospholipids, triglycerides, and cholesterol esters was clearly visible. FT-IR spectra of EVs and cells, from which they derived, differed in the 900–1300 cm^−1^ area that is typically referred to as the stretching vibration of the nucleic acid backbone. This could be reasonably ascribable to the consistently different content of RNA and DNA in EVs and cells [50,51].

The comparison of sensitive and doxorubicin-resistant OS cell-derived EVs revealed a similar protein organization, although with a greater variable protein-to-lipid ratio in EVs/DXR. These data are in line with recent literature that correlated lipid composition of EVs with EV release and cancer aggressiveness [52,53].

After proper normalization of the Amide I band at 1649 cm^−1^, peculiar characteristics in the profile of sensitive and doxorubicin-resistant cells-derived EVs become clearly visible. A shift in the band regions of 1650–1545 cm^−1^ and a band at 1601 cm^−1^, which characterize the sensitive EV population, enabled to unambiguously distinguish spectra of EVs/s and EVs/DXR. This could be ascribed to a modification of the Amide I band and of the α-helix conformation [54]. Indeed, in FTIR spectroscopy, different peak positions of the Amide I band in the 1600 to 1700 cm^−1^ area correlate to different secondary structures of the proteins (i.e., unstructured conformation vs. α-helix conformation) and can be used to monitor conformation changes in proteins/peptides [55,56].

The differences between FT-IR spectra of sensitive and resistant EVs were not detectable in the corresponding cells, and this can be reasonably associated with the higher level of complexity, and the number of different structures composing a cell, compared to an EV. In addition, relevant bands in the 800–1300 cm^−1^ area in the cells overlap with sharp bands of PBS and might be completely concealed, so proper identification of their position and intensity might be hampered, even after PBS subtraction. Thus, EVs represent a circulating tumor component with specific subgroup characteristics much more easily recognizable than cells by spectroscopy.

We further analyzed FT-IR spectra profiles of mesenchymal stromal cells, as a model of normal human cells, and MSC-derived EVs. Notably, MSC and OS cell FT-IR spectra profiles were similar, while OS and MSC-derived EV profiles and the spectroscopic protein-to-lipid ratio were completely different. These findings further strengthened the possibility to be able to distinguish among EVs populations by FT-IR spectroscopy.

Furthermore, the application of liquid biopsy approaches required the identification of reliable and easily detectable biomarkers, with affordable costs. It is noteworthy that only a few micrograms of EVs are sufficient to obtain clear and reliable FT-IR spectra signals. In addition, vibrational spectroscopy is label-free, sensitive, and requires minimal sample preparation. Moreover, machine-learning approaches can be used to analyze IR spectra in an automated fashion, which represents an additional advantage for future clinical applications.

## 5. Conclusions

In this study, we demonstrated the possibility to characterize OS-derived EVs by Fourier transform infrared spectroscopy (FT-IR) to define a specific spectral signature for drug-resistant OS-derived EVs. Since EVs can be considered circulating markers of cancer progression, aggressiveness, and resistance to therapy, our findings represent the starting point to explore the possible use of the FT-IR technique to identify EVs with an aggressive phenotype.

## Figures and Tables

**Figure 1 cells-11-00778-f001:**
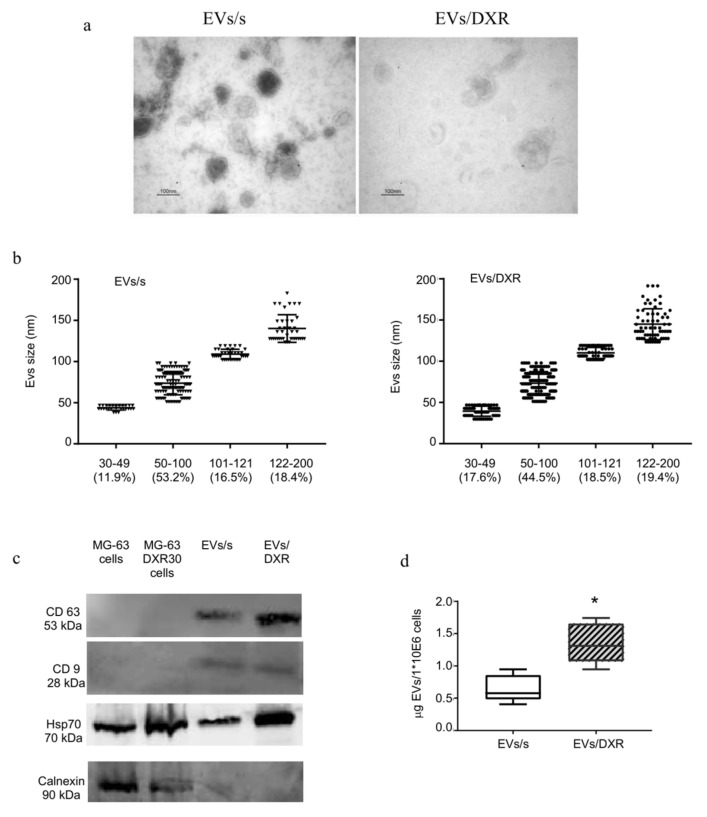
Osteosarcoma–derived extracellular nanovesicles (EVs) characterization. (**a**) Representative transmission electron microscopy images of EVs, isolated from medium conditioned by MG-63 (EVs/s) and MG-63DXR30 (EVs/DXR) cells (scale bar: 100 nm). (**b**) The distribution of vesicles in different size classes is described as percentage of distribution, with evidence of individual measurements, mean ± standard deviation. (**c**) Protein content-based EVs characterization was assessed by Western blot analysis for the expression of transmembrane proteins, cytosolic protein, and endoplasmic reticulum protein. (**d**) Box-plots depicting the release of EVs by OS cells, that was quantified by protein assay and normalized on 1 × 10^6^ viable cells (median and min-max values are shown, n = 5, Mann–Whitney test, * *p*-value = 0.04).

**Figure 2 cells-11-00778-f002:**
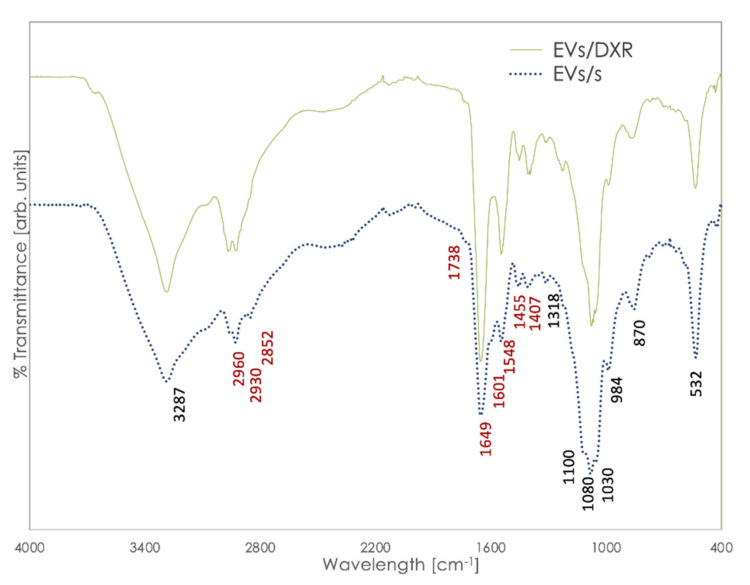
Representative spectra of EVs/s and EVs/DXR after subtracting PBS curve. Bands in red are those relevant to lipids and proteins structure, experiencing differences among different groups.

**Figure 3 cells-11-00778-f003:**
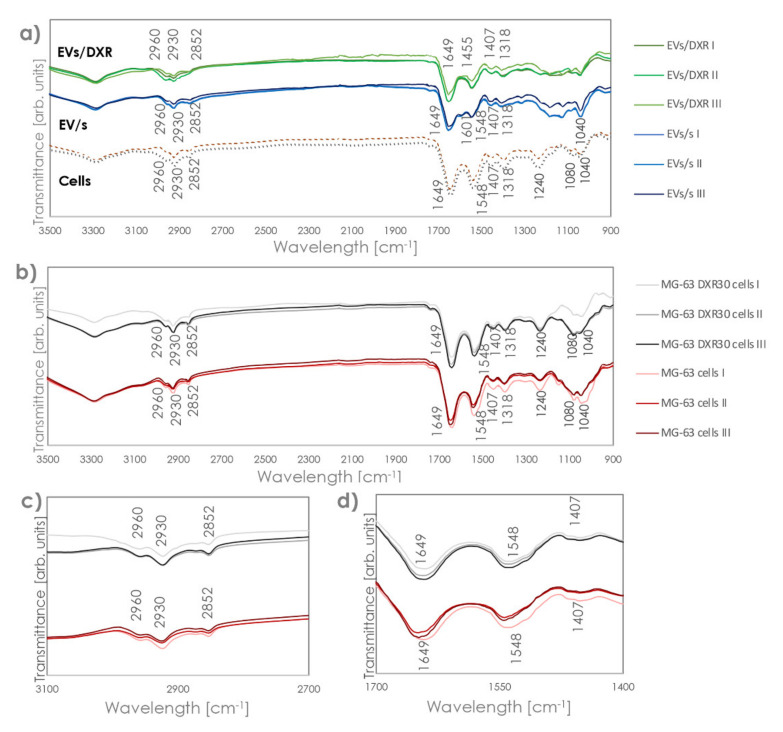
FT–IR/ATR spectra of EVs/s, EVs/DXR, and cells from which they are derived. (**a**) FT–IR spectra after subtracting the PBS reference curve of OS cells and OS derived–EVs. (**b**) FT–IR spectra of OS sensitive and drug-resistant cells. Panels (**c**,**d**) highlight the band region corresponding to protein and lipid zones, respectively.

**Figure 4 cells-11-00778-f004:**
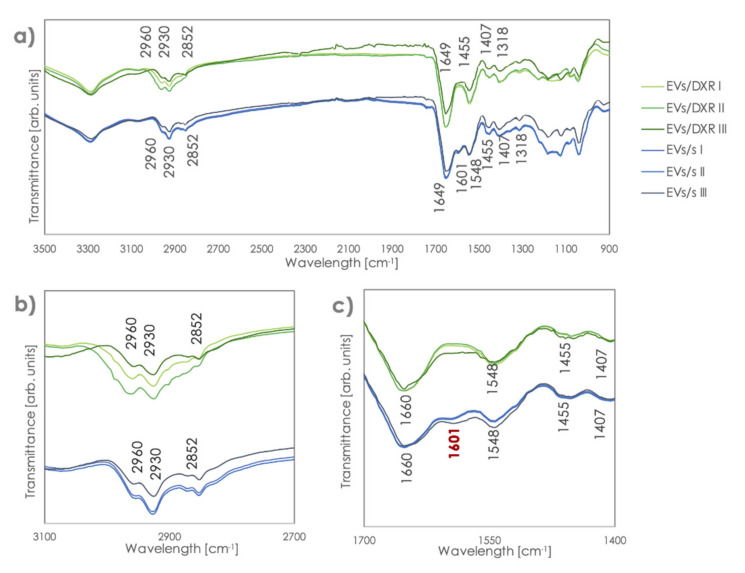
FT–IR/ATR spectra of EVs/s and EVs/DXR. (**a**) Spectra of EVs after scaling with respect to the band at 1649 cm^−1^ and zooms on the (**b**) lipid and (**c**) protein areas. The band in red, at 1601 cm^−1^, permits distinction between sensitive and resistant EVs.

**Figure 5 cells-11-00778-f005:**
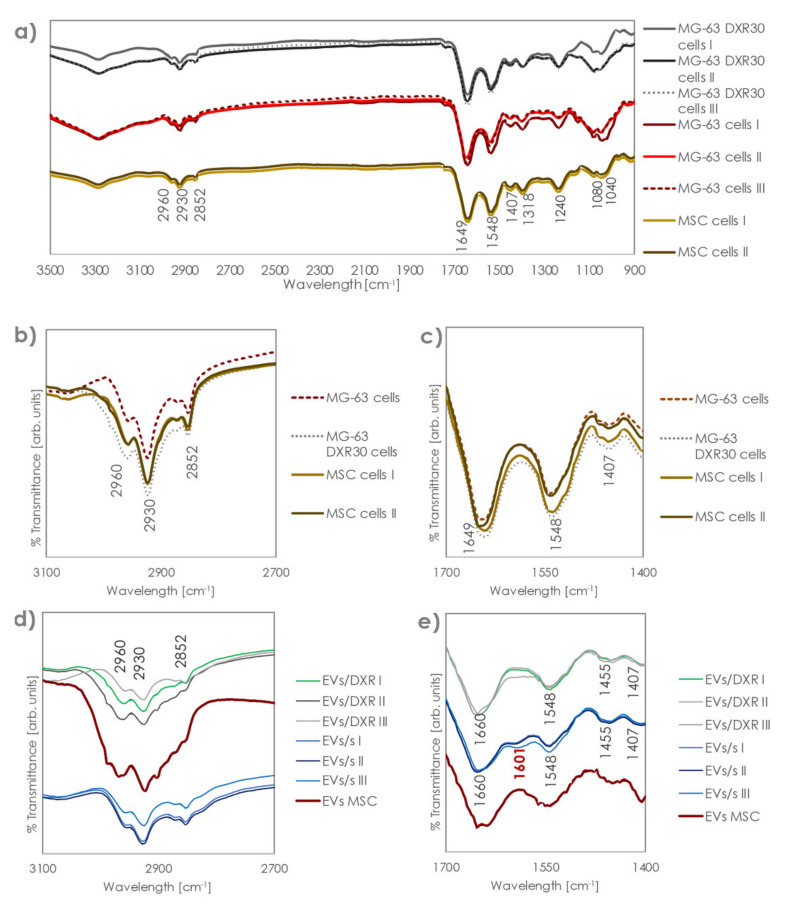
FT–IR/ATR spectra of OS EVs/s, OS EVs/DXR, MSC EVs, and cells from which they were derived. (**a**) FT–IR spectra after subtracting the PBS reference curve and zooms on the (**b**) lipid and (**c**) protein areas for OS and ADMSC cells. Panel (**d**,**e**) highlight the band region corresponding to protein and lipid zones, respectively, for EVs samples.

## Data Availability

Data are available upon request.

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
