# Peer review of "FT-IR Spectral Signature of Sensitive and Multidrug-Resistant Osteosarcoma Cell-Derived Extracellular Nanovesicles"

_cells, 2022, doi:10.3390/cells11050778_

Round 1
Reviewer 1 Report
Authors should ascribe FTIR peaks to their corresponding bands on the graphs for easy and quick grasping for readers
Author Response
Authors should ascribe FTIR peaks to their corresponding bands on the graphs for easy and quick grasping for readers
Figures 3 and 5 were modified adding the bands labels, as suggested by the Reviewer.
Reviewer 2 Report
In their manuscript entitled “FT-IR spectral signature of sensitive and multidrug resistant osteosarcoma cell-derived extracellular nanovesicles”, Perut and colleagues describe the application of fourier-transform infrared spectroscopy (FT-IR) to distinguish extracellular vesicles (EV) derived from a chemoresistant osteosarcoma cell line and EVs from a chemosensitive osteosarcoma cell line. The authors identify a specific signature to distinguish between EVs originating from the different osteosarcoma cells, among others characterized by a band at 1601 cm-1. The authors conclude that their findings might be useful to monitor the progression and therapy resistance of osteosarcoma non-invasively. The manuscript is written understandable and logical. The methodology is appropriate and well explained and the data seems sound. However, I have some concerns on this manuscript.
Here are my concerns in detail:
1) The biggest weakness of this study is that only two cell lines (merely one cell line and its chemo-resistant descendant) were used. Therefore it is hard to assess the generality of the study – is the band detected at 1601cm-1 a common feature of EVs from chemosensitive osteosarcoma cells/cell lines or specific to MG-63? And does it exist in the EVs of other chemosensitive tumor cells as well, or is a common feature of osteosarcoma? To be fair, although the results are interesting, to my opinion the experiments are too narrow to justify publication on this stage, especially as it is not a proof-of-principle study.
2) Figure 1b: I find it interesting, that the EVs shedded by the DXR OS cells seem to be in average smaller than the EVs from their chemotherapy sensitive counterparts. Did the authors analyze whether this was a statistical significant effect? If yes, is there an explanation for these seemingly smaller EVs?
3) Figure 1c: CD 81 is not shown here, is there a reason for? In the original blot, there is barely no signal? Furthermore, do the authors have an explanation for the observation, that there are no CD9 and CD63 bands visible in the OS cells Western-Blot?
4) Figure 5a: no MSCs included here?
5) I presume the explanation given in l. 350-351 on the nature of the band shifts and the 1601 cm-1 band rather scarce. I would recommend exemplifying in some more sentences and maybe – if available - inserting some more references.
6) I would recommend to insert one or two more references in l. 46 additionally to [5]
7) Please state the meaning of the different number colors (red and black) in Figure 2 and 4.
Finally, I want to thank the authors for sharing their results with the scientific community and apologize for the additional workload. Best regards.
Author Response
1) The biggest weakness of this study is that only two cell lines (merely one cell line and its chemo-resistant descendant) were used. Therefore it is hard to assess the generality of the study – is the band detected at 1601cm-1 a common feature of EVs from chemosensitive osteosarcoma cells/cell lines or specific to MG-63? And does it exist in the EVs of other chemosensitive tumor cells as well, or is a common feature of osteosarcoma? To be fair, although the results are interesting, to my opinion the experiments are too narrow to justify publication on this stage, especially as it is not a proof-of-principle study.
According to the reviewer suggestions, we analyzed FT-IR spectra EVs in an additional chemosensitive OS cell line (i.e. 143B cells), and we find that the band detected at 1601cm-1 in MG63 is also present in EVs derived from 143B cells. This data adds further credence to the concept that the band detected at 1601cm-1 is a common feature of EVs from chemosensitive osteosarcoma cells and not specific to MG-63. This finding was added in the result section (supplementary Figure S3).
We agree that it could be very interesting to explore FT-IR spectra characteristics in other tumor derived chemosensitive and resistant EVs, but this will be the object of future studies.
2) Figure 1b: I find it interesting, that the EVs shedded by the DXR OS cells seem to be in average smaller than the EVs from their chemotherapy sensitive counterparts. Did the authors analyze whether this was a statistical significant effect? If yes, is there an explanation for these seemingly smaller EVs?
According to the reviewer suggestions, we verified if there is any statistical difference in size between sensitive and corresponding resistant cells derived EVs. There was no significant difference in EVs size distribution. Our results are similar to Peak et al. findings in sensitive and drug resistant prostate cancer derived EVs.
- Peak, T.C.; Panigrahi, G.K.; Praharaj, P.P.; Su, Y.; Shi, L.; Chyr, J.; Rivera-Chávez, J.; Flores-Bocanegra, L.; Singh, R.; Vander Griend, D.J.; et al. Syntaxin 6-mediated exosome secretion regulates enzalutamide resistance in prostate cancer. Mol. Carcinog. 2020, 59, 62-72. doi: 10.1002/mc.23129.
3) Figure 1c: CD 81 is not shown here, is there a reason for? In the original blot, there is barely no signal? Furthermore, do the authors have an explanation for the observation, that there are no CD9 and CD63 bands visible in the OS cells Western-Blot?
CD81 was nearly undetectable, as it was detected only upon extended exposure of the membrane. The very low detection of CD81 in MG63- derived EVs was assessed also by Jerez et al (2017).
An accumulation of CD9 and CD63 in small EVs as compared to whole cell lysates is usually detected. Probably, we did not observed clearly distinguishable/visible CD9 and CD63 bands in our samples (Fig 1C) as we did not exposed our blot for long time.
- Jerez S et al. Proteomic Analysis of Exosomes and Exosome-Free Conditioned Media From Human Osteosarcoma Cell Lines Reveals Secretion of Proteins Related to Tumor Progression. J Cell Biochem. 2017 Feb;118(2):351-360.doi: 10.1002/jcb.25642. Epub 2016 Aug 15.
4) Figure 5a: no MSCs included here?
According to the reviewer suggestions, we included MSC FT-IR profiles in Fig 5a (Fig 5a rev).
5) I presume the explanation given in l. 350-351 on the nature of the band shifts and the 1601 cm-1 band rather scarce. I would recommend exemplifying in some more sentences and maybe – if available - inserting some more references.
This point was better clarified, as in the following: “Indeed, in FTIR spectroscopy, different peak positions of the Amide I bands in the 1600 to 1700 cm−1 area correlate to different secondary structures of the proteins (i.e. unstructured conformation vs α-helix conformation) and can be used to monitor conformation changes in proteins/peptides. [Wang C et al, 2010; Dziri et al, 1999].” More references were added to support the evidence.
- Wang, C.; Shah, N.; Thakur, G.; Zhou, F.; Leblanc, R.M. Alpha-synuclein in alpha-helical conformation at air-water interface: implication of conformation and orientation changes during its accumulation/aggregation. Chem. Commun. 2010, 46, 6702-6704. doi: 10.1039/c0cc02098b.
- Dziri, L.; Desbat, B.; Leblanc, R.M. Polarization-Modulated FT-IR Spectroscopy Studies of Acetylcholinesterase Secondary Structure at the Air−Water Interface. J. Am. Chem. Soc. 1999, 41, 9618–9625. doi: 10.1021/ja990099d
6) I would recommend to insert one or two more references in l. 46 additionally to [5]
According to reviewer suggestions we added these references:
- Chen, R.; Wang, G.; Zheng,Y.; Hua, Y.; Cai, Z. Drug resistance-related microRNAs in osteosarcoma: translating basic evidence into therapeutic strategies. J. Cell. Mol. Med. 2019, 23, 280–292. doi: 10.1111/jcmm.1406.
- Torreggiani, E.; Roncuzzi, L.; Perut, F.; Zini, N.; Baldini, N. Multimodal transfer of MDR by exosomes in human osteosarcoma. J. Oncol. 2016, 49, 189-196. doi: 10.3892/ijo.2016.3509
- De Martino, V.; Rossi, M.; Battafarano, G.; Pepe, J.; Minisola, S.; Del Fattore, A. Extracellular Vesicles in Osteosarcoma: Antagonists or Therapeutic Agents? J. Mol. Sci. 2021, 22, 12586. doi:10.3390/ijms222212586.
7) Please state the meaning of the different number colors (red and black) in Figure 2 and 4.
Bands in red are those that define differences among the different groups. This was explained in the relevant captions.
Reviewer 3 Report
The submitted manuscript focuses on the analysis of EVs derived from sensitive and multi-drug-resistant osteosarcoma cells with FT-IR spectroscopy after ultracentrifugation. Beside size, morphology and specific protein expression, EVs were further characterized and defined by a specific spectral signature by FT-IR spectroscopy. That method opens an additional approach to the other applied techniques; however, as the underlying alterations behind the different spectral wavelengths (in lipids or proteins) would always need further analysis, I am not sure if the FT-IR spectroscopy adds as significant value to the established methods. I have the impression that it is best suited for a prescreening of EV populations before deeper analysis.
Minor concerns:
1. Were the EVs used for TEM also frozen on -80 before use?
2. How many replicates were peformed for the experiments? n=5?
3. Can EV quantity/number be determined by the Bradford assay? This would assume that all EVs have an identical protein content (cargo and membrane). I would suggest to do a nanoparticle counting.
4. In the supplement figure with the original western blots I miss a molecular weight marker, several lanes are not indicated.
5. Can FT-IR spectroscopy distinguish between EVs and cell fragments/proteins?
Author Response
- Were the EVs used for TEM also frozen on -80 before use?
In all experiments we used EVs previously frozen at -80°C.
- How many replicates were peformed for the experiments? n=5?
The quantification of EVs, in terms of protein content, was performed in 5 different samples (n=5), and we specified it in Fig.1 D legend. We evaluated specific FT-IR profiles in three different samples of sensitive and chemoresistant MG63 cells and derived EVs, as shown in Fig 3-4.
- Can EV quantity/number be determined by the Bradford assay? This would assume that all EVs have an identical protein content (cargo and membrane). I would suggest to do a nanoparticle counting.
We agree with the reviewer comment. Indeed, according to MISEV 2018 position paper (Thery al, 2018), different methods can be used to quantify EVs. MISEV 2018 recommends that each preparation of EVs be defined by quantitative measures of the source of EVs (e.g. number of secreting cells, volume of bio- fluid, mass of tissue) and characterized to determine abundance of EVs (total particle number and/or protein or lipid content). The choice of the quantification methods is linked to the aim of a specific study, as EVs analysis is severely hampered by the EVs heterogeneity, and all the most commonly applied methods showed limitations (Harties TA, 2019).
In this study, we demonstrated that drug resistant OS cells release significantly higher amounts of EVs compared to the respective sensitive cells, in terms of total protein content. In prostate cancer, a higher amount of EVs release by drug resistant cells was assessed in terms of total protein content (Henrich Se et, al, 2020) and, similarly, by NTA analysis (Peak et al. 2020).
- Thery C. et al., Minimal information for studies of extracellular vesicles 2018 (MISEV2018): a position statement of the International Society for Extracellular Vesicles and update of the MISEV2014 guidelines. J Extracell Vesicles. 2018 Nov 23;7(1):1535750. doi: 10.1080/20013078.2018.1535750. eCollection 2018.
- Henrich SE et al. Prostate cancer extracellular vesicles mediate intercellular communication with bone marrow cells and promote metastasis in a cholesterol‐dependent manner. Extracell. Vesicles. 2020, 10, e12042. doi: 10.1002/jev2.12042.
- Peak TC et al. Syntaxin 6-mediated exosome secretion regulates enzalutamide resistance in prostate cancer. Carcinog. 2020, 59, 62-72. doi: 10.1002/mc.23129.
- In the supplement figure with the original western blots I miss a molecular weight marker, several lanes are not indicated.
According to the reviewer suggestions, we added additional original images of western blots and we indicated molecular weight markers.
- Can FT-IR spectroscopy distinguish between EVs and cell fragments/proteins?
FT-IR can distinguish among different classes of compounds only if the differences cause a modification in the chemical bonds that characterize them. As a consequence, in principle, cells fragments can be clearly distinguished from the EVs, while proteins inside or outside the EVs cannot be distinguished. However, the possibility to detect the relevant modifications in the bands also depends on their extent, hence it must be assessed case by case. Thus, the reviewer raises an interesting question, which could be addressed in a proper study with a different experimental setting.
Round 2
Reviewer 2 Report
In their manuscript entitled “FT-IR spectral signature of sensitive and multidrug resistant osteosarcoma cell-derived extracellular nanovesicles”, Perut and colleagues describe the application of fourier-transform infrared spectroscopy (FT-IR) to distinguish extracellular vesicles (EV) derived from a chemoresistant osteosarcoma cell line and EVs from a chemosensitive osteosarcoma cell line. The authors identify a specific signature to distinguish between EVs originating from the different osteosarcoma cells, among others characterized by a band at 1601 cm-1. The authors conclude that their findings might be useful to monitor the progression and therapy resistance of osteosarcoma non-invasively. This is the first revision of the manuscript.
The authors responded to all my remarks to my full satisfaction. I do not have any other points and thank the authors for the clarifications induced in the manuscript.
Best regards.